# Application of Nanopore Sequencing (MinION) for the Analysis of Bacteriome and Resistome of Bean Sprouts

**DOI:** 10.3390/microorganisms9050937

**Published:** 2021-04-27

**Authors:** Milada Solcova, Katerina Demnerova, Sabina Purkrtova

**Affiliations:** Department of Biochemistry and Microbiology, Faculty of Food and Biochemical Technology, University of Chemistry and Technology, Technicka 3, 16628 Prague 6, Czech Republic; Katerina.Demnerova@vscht.cz

**Keywords:** spread of antibiotic resistance, antibiotic resistance genes, food chain, food safety, nanopore sequencing, MinION, MALDI-TOF MS, bacteriome, resistome, bean sprouts

## Abstract

The aspiration these days is to apply rapid methods for parallel analysis of bacteriome and resistome of food samples to increase food safety and prevent antibiotic resistance genes (ARGs) spreading. In this work, we used nanopore sequencing (NS) to determine the diversity and dynamics of the microbiome and resistome in two types of bean sprouts. We proved that NS provided an easy, quick, and reliable way to identify the microbiome and resistome of a food sample also. The species diversity obtained by NS and by cultivation methods with MALDI-TOF MS identification was comparable. In both samples, before and after cultivation (30 °C, 48 h), the dominant part of bacteriome formed *Gammaproteobacteria* (*Enterobacteriaceae, Erwiniaceae, Pseudomonadaceae, Moraxellaceae*) and then *Firmicutes* (*Streptococcaceae*). The diversity and abundance of single ARGs groups were comparable for both samples despite bacteriome differences. More than 50% of the detected ARGs alignments were mutations conferring resistance to aminoglycosides (16S rRNA), resistance to fluoroquinolones (*gyr*A, *gyr*B, *par*C, *par*D) and elfamycin (EF-Tu). ARGs encoding efflux pumps formed more than 30% of the detected alignments. Beta-lactamases were represented by many variants, but were less abundant.

## 1. Introduction

The spread of antibiotic resistance has become a serious problem. Antibiotic resistance has adverse effects not only on human, but also on livestock, health care, agriculture, and the environment. The current trend to reduce the incidence of antibiotic-resistant bacteria (ARB) is therefore the supervision of the appropriate use of antibiotics (legislation, dissemination of information on this issue) and the development of new methods to improve the detection of ARB and especially genes encoding antibiotic resistance genes (ARGs) [1]. ARB and ARGs can be found in the environment (soil, water), human or animal feces, food, and the gastrointestinal tract, which can serve as their reservoirs (hot spots). Bacteria can transmit ARGs through horizontal gene transfer (HGT) and further maintain them in bacterial populations, creating multiresistant strains through genetic structures such as plasmids, integrons, and transposons [2,3,4,5]. The food can be contaminated with ARB and ARGs in several ways. ARB may be present in primary food products (raw meat, milk, fermented dairy products) [6] due to the usage of antibiotics (ATB) in agricultural production (e.g., animal medications) or using ARB contaminated water for irrigation or fertilizers [7]. Cross-contamination with ARB also may occur during food processing (unsanitary food handling, contaminated equipment, pathogens from the human reservoir, rodents, and insects) [7,8]. ARB can also be present as starter cultures, bioprobiotics, and biopreserved microorganisms [7]. ARGs can then circulate through the food chain, increasing the amount of ARGs in pathogenic bacteria but also in the human bacteriome [9,10].

Phenotypic methods (disk diffusion method, E-test, broth microdilution), standard and widely used for the detection of ARB in laboratories, are time and material consuming [11]. For speeding-up and higher accuracy, they use genotypic molecular methods (PCR, DNA microarray), which detect the presence or absence of a given ARG directly or quantify it [12].

The identification of ARB can be performed through phenotypic biochemical testing or protein analysis such as MALDI-TOF MS (Matrix-Assisted Laser Desorption Ionization Time of Flight Mass Spectrometry) or very reliable, but more time requiring 16S rRNA or other gene sequencing. MALDI-TOF MS is a widely accepted and reliable method. It is a very rapid and easy method in comparison with any other phenotype or genotype identification methods. However, it also has some limitations such as the lack of protein profile of some species in the database and the higher cost of MALDI-TOF MS system [13]. 

Over the last decade, significant progress in sequencing technologies arose, resulting in extremely efficient and powerful technologies, referred to as the next generation sequencing (NGS) methods [1]. NGS techniques were introduced to the laboratory practice in the early 2000s [14]. Generally, NGS techniques can be used for whole genome sequencing of a single strain or strain mixture or sequencing DNA isolated from a particular sample (metagenomic analysis) [15]. With NGS techniques it is possible not only to detect the source and transmission of antimicrobial contamination [16], including food-borne pathogens, but also to track the spreading and transmission of ARB and ARGs or to characterize the epidemic spread of bacterial pathogens [17,18]. The NGS methods are used not only to identify already known ARGs and their genetic carriers (plasmids) and to monitor multidrug-resistant strains, but also to predict new ARGs and their variants [1].

These days, two NGS methods often used are Illumina Solexa (short reads) and nanopore sequencing (long reads) [2,15]. Illumina Solexa dominates the second generation of NGS, requiring amplification of the template (to fragments), which may result in errors or loss of information. Illumina Solexa sequencing runs produce in individual spots for a large number of single-stranded copies of the target sequence, which are then read in parallel by using fluorescently labelled nucleotides [2,19].

The principle of third-generation NGS is the sequencing of individual DNA molecules without their prior amplification and the parallel sequencing of many copies. The third-generation NGS involves several different methods such as Single-Molecule Real-Time Sequencing (SMRT) or PacBio RSII system from Pacific Biosciences or NS [2,20]. 

NS is based on the detection of changes in the electrical conductivity that occurs when a strand of DNA passes through a nanopore situated in a flow cell. These changes are specific for each base due to its different electrical resistance. The NS system was developed 2014 by Oxford Nanopore Technologies and it can run in diverse platforms (MinION, Gridion and others), which differ in the amount and yield per flow cell [21]. The sequencing MinION platform with dimensions 105 × 23 × 23 mm can be connected to the standard USB port on any computer. The analysis of sequencing data can be performed automatically using EPI2ME software (covering different applications such as, e.g., ARGs detection or 16S rRNA gene sequencing) or manually, also in real time during the actual measurement. Mobility and user-friendly software for real-time data analysis have made the MinION particularly attractive for clinical diagnostics in the field. The cost of each analysis is additionally compared significantly with previous systems and can be decreased by using barcoding [2,22].

The tendency of the present day is to develop and introduce new methods for the detection, determination, or yet quantification of ARGs. Our goal was to establish an efficient and appropriate protocol for NS to detect the occurrence, abundance, and species distribution of ARGs (resistome) in mungo sprouts including the global microbial species distribution (bacteriome). Generally, examination of the resistome and bacteriome of a given food sample using NS is an extremely promising method for quick and very complex analysis of food samples as well as for many other microbiologically analyzed samples [23].

The life cycle of ARGs includes all parts of the environments and human activities, including the introduction of ARGs to the human microbiome through food. Besides the fact that food can bear pathogenic ARB, also the commensal or conditionally pathogenic bacteria bearing ARGs can influence human health. It was found that consumed food has an impact on the diversity of human bacteriome [24]. The most problematic area for bacteria transfer is food consumed raw, without heating. One of such foods are sprouts.

Mungo bean and cereal sprouts were chosen for this method implementation for more reasons. They are known from previous analyses (data not shown) to contain quite high loads of microorganisms and also due their high potential for microbial contamination (a difficult journey from farm to consumer). As they are high in raw consumption (preservation of all nutrients), they can serve as an excellent foyer for ARB into the human gastrointestinal microbiome, and their consumption can lead to food poisoning [25,26].

The raw sprouts are known to often be connected with *Salmonella* spp. and *E. coli* outbreaks [25,27]. From 2000 to 2002, seven outbreaks of *Salmonella enteritidis* associated with sprouts were identified in the United States, Canada, and Netherland [28]. *Salmonella enterica* serovar Bareilly (outbreak of 190 illnesses) was detected in bean sprouts (2010, UK) and in fenugreek sprouts (2011, Germany) was detected *E. coli* O104:H4 (3842 illnesses with 53 deaths) [25].

The risk factors for high microbial contamination are categorized as preharvest contamination (type of used fertilizer, irrigation water, or soil quality) [29,30] and postharvest contamination (transportation, storage of sprout, handling, and hygiene of workers) [25]. Attachment of bacteria (including possible foodborne pathogens) to the seed surface, which supports bacterial survival, is enabled mainly by cavities in the seeds, when the attachment may occur through the fimbriae, flagella, and biofilms [25]. These factors can render sprouts to be an important vehicle for spreading ARGs between different milieus. 

Two samples of different sprouts (mungo bean sprouts and a mixture of legume and cereal sprout) were used to implement the selected methodologies. Specifically, these protocols introduced a NS method to determine the diversity and dynamics of microbiome and resistome in sprouts and compared this method with the results obtained by cultivation methods (including identification by MALDI-TOF MS).

## 2. Materials and Methods

### 2.1. Study Design

An approach to monitoring the occurrence of ARGs in mungo sprouts was designed. Sprouts samples, mixed with buffered peptone water (BPW) (1:10), were analyzed twice (before and after cultivation). The plating methods were used to count the target groups of microorganisms and to isolate them. The isolates were identified by MALDI-TOF MS method. DNA was isolated from 1 mL of sample suspension by ChargeSwitch Nucleic Acid Purification Technology kit and then sequenced by the NS platform MinION (Oxford Nanopore Technologies) and analyzed by software EPI2ME. 

### 2.2. Samples

In this study, two different samples of sprouts were analyzed, both bought in a Czech Republic retail market in 2020. Sample A by Czech Republic producer was bought in January 2020. Sample B by German producer was bought in May 2020. Sample A contained only mungo bean sprouts (*Vigna radiata*), with seed origin from Barma (Myanmar). Sample B was a mix of legumes and cereal sprouts from bioecological agriculture (mungo sprouts 50%, wheat sprouts (*Triticum turgidum*) 25%, lentil sprouts (*Lens culinaris*) 25%). The origin of seeds in sample B was not defined. 

### 2.3. Cultivation Methods

The amount of 25 g of food sample was mixed with 225 mL of buffered peptone water (BPW, HiMedia, Mumbai, India) and stomachered for one minute. Before and after cultivation (30 °C, 48 h), plating colony count methods for target microorganisms on media and cultivation conditions by the appropriate ISO methods were performed. The mixture was decimally diluted in sterile saline solution. The enumeration by pour plate method (1 mL) was done for: colony count at 30 °C by the pour plate technique on Plate Count Agar (HiMedia, India) (PCA, 30 °C, 72 ± 3 h, ISO 4833), *Enterobacteriaceae* on Violet Red Bile Dextrose agar (Merck, Germany) (VRBD, 30 °C, 72 ± 3 h, ISO 21528-2), *E. coli* on Tryptone Bile X-glucuronide agar TBX (Merck, Darmstadt, Germany) (TBX, 44 °C, 18–24 h, ISO 16649-2) and lactic acid bacteria on de Man, Rogosa and Sharpe agar (Merck, Darmstadt, Germany) (MRS, 30 °C, 48 h, ISO 15214). The enumeration by spread plate method (0.1 mL) were done for: presumptive *Bacillus cereus* on Mannitol egg yolk polymycin agar (Merck, Darmstadt, Germany) (MYP, 30 °C, 24–48 h, ISO 7932), coagulase positive staphylococci on Baird-Parker agar (Merck, Darmstadt, Germany) (BP, 37 °C, 48 h ISO 6888-1), yeast and fungi on Dichloran-rose bengal chloramphenicol agar (Merck, Darmstadt, Germany) (DRBC, 25 °C, 72 h, ISO 21527). Non-ISO based spread plate method using Columbia agar with 5% sheep blood (BioRad, Hercules, CA, USA) (CSB, 30 °C, 48 h) was used to enumerate and isolate a broad range of non and more fastidious bacterial species. The quality control of the media was performed according to ČSN ISO 11133. All plate count analysis were performed in duplicates. 

### 2.4. Identification and Isolation of Bacteria

After cultivation from every plate of different media and single dilutions, all different types of colonies in replicates were randomly subisolated according to their abundancy on CSB (30 °C, 48 h). These isolates were later identified by MALDI-TOF MS (Matrix-Assisted Laser Desorption Ionization Time of Flight Mass Spectrometry) with Autoflex Speed mass spectrometer (Bruker Daltonics, Bremen, Germany) and database Biotyper 3.1 (Bruker Daltonics, Bremen, Germany).

The matrix was the solution of alpha-cyano-4-hydroxycinnamic (HCCA) acid (Bruker Daltonics, Bremen, Germany) at a concentration of 10 mg·ml^−1^ in organic solvent. One ml organic solvent was prepared from 500 μL acetonitrile, 250 μL 10% trifluoroacetic acid and 250 μL nuclease free water (ThermoFisher, Waltham, MA, USA). The samples were spotted on a polished steel plate MTP 384 (Bruker Daltonics, Bremen, Germany). 

Three methods for sample preparation were used, which differ in the efficiency of extraction of intracellular proteins and in the time consumption. The first-choice method was eDT and the required limit for probable species identification was score value 2.0.

The direct transfer (DT) method is based on the direct application of the tested culture on spots of the steel MALDI plate. In the eDT (extended direct transfer) method, after application of the culture on spots, the sample is first covered with 2 μL of 70% solution of formic acid. The extraction method (Ex EtOh/FA) uses ethanol and 70% formic acid to disrupt the cell wall of microorganisms and releases intracellular proteins into a solution of ethanol and formic acid. For the extraction method, 300 μL of microbial suspension in sterile distilled water was mixed with 900 μL of absolute ethanol, after centrifugation (2 min, 13,000 rpm) the pellet was resuspended in 50 μL of 70% formic acid and then 50 μL of acetonitrile). After centrifugation (2 min, 13,000 rpm), 2 μL of supernatant was applied to the spots and left to dry. All spots were then covered with HCCA matrix and allowed to dry for at least 15 min at room temperature, when they turned yellow. For the calibration, Bruker Bacterial Test Standard (Bruker BTS) was used. All chemicals were purchased from Sigma-Aldrich (St. Louis, MO, USA) except nuclease free water (ThermoFisher, Waltham, MA, USA).

### 2.5. DNA Isolation and Quantification

The total DNA was isolated from 1 mL of the mixture of sprouts and buffered peptone water (before and after cultivation) by ChargeSwitch Nucleic Acid Purification Technology (ThermoFisher, Waltham, MA, USA) according to the producer’s instructions. The principle is to use charged magnetic beads, which charge depends on the pH of the surrounding buffer. 

The concentration and purity of DNA was measured by Nanodrop Implen NanoPhotometer (ThermoFisher, Waltham, MA, USA). The DNA quality parameters (ratio A260/A280 of 1.8 and A260/230 of 2.0–2.2) affect the quality of the sequenced data. 

DNA isolation always includes 1 to 1.5 h of sample preparation (incubation). Besides this, for 1 sample another 15 min of processing is required, which is shortened for 12 samples to 2 h.

### 2.6. Nanopore Sequencing

The library was prepared with Ligation-sequencing kit (SQK-LSK109) (Oxford Nanopore Technologies, Oxford, United Kingdom). The sequencing run was performed in flow cell R9.4.1 with MinION device (Oxford Nanopore Technologies, Oxford, United Kingdom). The recommended input DNA was 1000 ng for this kit and flow cell R9.4.1. In this experiment, 100 ng (because of low concentration of DNA) was used per sample before cultivation and 1000 ng per sample after cultivation. 

The steps for the basic ligation sequencing kit are as follows. The first step is to prepare a library in which the fragmented DNA is repaired, and their ends are prepared (35 min) for the ligation of adapters and purified (30 min). The second step is to load the sample (10 min) onto the flow cell (with nanopores) settled in the sequencing device, to set up in MinKNOW software for the sequencing run and to start sequencing [22]. Leading adapters navigate DNA fragments into the open nanopore channel and the DNA strand grabbing through the nanopore channel is facilitated by other adapters (hairpin and end adapters) [31]. The passage of both complementary strands through the pores ensures higher accuracy [32]. 

Preparation of the DNA library consists of DNA repair and end-prep (preparation of DNA in nuclease free water, preparation of a mixture of DNA with consumables (NEBNext FFPE Repair Mix and NEBNext End repair/dA-tailing Module) to prepare DNA ends for binding adapters and AMPure XP bead clean-up), adapter ligation (preparation of mixture of DNA with ligation buffer and ligase (NEBNext Quick Ligation Module)), AMPure XP bead clean-up, priming (mixture of sequencing buffer, loading beads and DNA library) and loading the SpotON flow cell followed by producer instructions. The consumables were purchased from New England Biolabs (Ipswich MA, USA). 

Internally calculated to prepare 1 sample and to run sequencing in a flow cell for 8 h, it costs 268 euros and brings approximately 533 Mbases with 311,467 reads (with the usage of a barcoding kit it is possible to reduce the price by 80%). 

Library preparation run required 1 h and 15 min for 1 sample. Nanopore sequencing runs up to 15 h. However, as the results by EPI2ME run in real-time, the first results are available after approximately 2 h and the complete results after 20 h (depending on the used PC capacity). Therefore, the complete analysis (from DNA isolation to EPI2ME results) requires 24 h for 1 sample or 30 h for 12 samples, if barcoding is used.

### 2.7. Sequencing and Data Analysis

For sequencing, MinKNOW software was used, which collects the sequencing data and converts it into basecalled reads. The sequencing run was 6 h for the sample before cultivation and 15 h for the sample after cultivation.

For the post-basecalling analysis, EPI2ME software (https://www.metrichor.com/, 25 January 2021) with WIMP (quantitative analysis tool for real-time species identification—a part of EPI2ME) was used for the evaluation of species diversity and antibiotic resistance genes distribution. EPI2ME software analyses sequenced data (fastq format) and its direct connection with databases (NCBI for species identification, CARD for ARGs identification). This approach thus simplifies to essential bioinformatics analysis. However, the provided DNA sequences and the raw signal used for basecalling enables other kinds of analysis if needed [31,32]. EPI2ME uses the laboratory and analysis workflow WIMP (What’s in my Pot?). WIMP filters fastq files with a mean q-score below a minimal threshold and the centrifuge classification engine assigns each read to a taxon in the NCBI taxonomy and reference database (RefSeq: NCBI reference sequence database) (https://www.ncbi.nlm.nih.gov/refseq/, 25 January 2021) [33,34].

EPI2ME uses CARD database (https://card.mcmaster.ca/, 17 February 2021) with molecular sequences of over 1600 known ARGs for the determination of antibiotic resistance [35]. CARD also includes a computer-generated study about the prevalence of these ARGs in 221 important pathogens (including sequence variants beyond those reported in the scientific literature). These sequences are products of active and ongoing curation of sequences available in GenBank with the usage of BLAST (basic local alignment search tool) and RGI (resistance gene identifier). The CARD database also includes data about the description of antibiotics and their targets along ARGs, associated proteins and antibiotic resistance literature. The main part of this database is ARO (antibiotic resistance ontology), which is important for the classification of ARGs data [35,36,37]. 

## 3. Results and Discussion

### 3.1. Nanopore Sequencing by Platform MinION (Oxford Nanopore Technologies, Oxford, UK)

The number and classification of reads obtained by nanopore sequencing totally and for single kingdoms are summarized in Table 1. 

For samples A and B before cultivation (0 h), 14,500 and 24,000 reads were obtained by analyzing 100 ng of isolated DNA. As the concentration of isolated DNA was 4.7 ng·µL^−1^ and 3.2 ng·µL^−1^, 100 ng was the maximum amount applicable for sequencing and it corresponded to 11%, resp. 16% DNA isolated from 1 mL of initial suspension with 0.1 g of sample. After 48 h of cultivation, 248,000, resp. 584,000 reads were obtained by analyzing 1000 ng DNA as the concentration of isolated DNA increased by 1.6 log for both samples (194 ng·µL^−1^ and 133 ng·µL^−1^). These reads corresponded to 3%, resp. 4% DNA isolated from 1 mL of suspension after cultivation with 0.1 g of sample. 

The level of read classification was significantly higher for sample A (86% 0 h, 85% 48 h) than for sample B (57% 0 h, 65% 48 h). As the percentages of unclassified reads were comparable before and after cultivation (for sample A: 14%, resp., 15%; for sample B: 43% and 35%), it should be supposed that sample B contained more cultivable, but not database identifiable species.

The classified reads comprise from the reads reliably assigned to some genus or species (classified at least genus level) and from reads with reliable assignment only to the higher ranks of the taxonomy tree (classified only to root). This only to root classification decreased in both samples after cultivation (from 29% to 2% and from 10% to 2%). It could be explained that before cultivation the samples could contain more dead cell DNA, which is supposed to be more fragmented, which complicates species identification.

The unclassified reads did not reach the required level of alignment (NCBI database), which is quantified by the mean q-score (quality score for ONT data, default minimum threshold 6) [22]. Generally, this mean q-score is influenced by the sequencing errors, the length of reads, and the range of the used reference database.

The viral and archaebacterial DNA formed maximally 1% of all classified reads in all cases, while eukaryotic DNA (DNA of protozoa, yeasts, and molds, but also human DNA) formed maximally 1% in all cases except sample B at 0 h (3%). Greninger et al. [38] used NS for the detection of three viruses from four human blood samples and later Ji et al. [39] detected RNA and DNA viruses in water samples. It shows the broad applicability of nanopore sequencing for analyzing different kingdoms of food microbiome in the case of interest and by using the most appropriate method for DNA isolation.

The bacterial alignments were formed at 97% (0 h) or 99% (48 h) of all classified reads and they were dominant. In both samples, the dominant phylum of uncultured samples (before cultivation, 0 h) was *Proteobacteria* (sample A 91.9% and sample B 97%), followed by *Firmicutes* (sample A 7.9% and sample B 2.6%), while *Bacteroidetes* were more or less frequent (in both samples under 0.5%). This corresponds partly to the findings of Margot et al. [26], who analyzed the uncultured bacteriome of mungo bean sprouts by 16S rRNA sequencing. Margot et al. also found *Proteobacteria* as the most prevalent phylum (90.4%), but *Firmicutes* (0.6%) were exceeded by *Bacteroidetes* (8.8%) [26]. Margot et al. observed that cultivation in BPW (37 °C and 42 °C, 4–8 h) selectively increases *Firmicutes* at the expense of *Proteobacteria*. We observed the same effect for BPW, 48 h at 30 °C in sample B, in which *Firmicutes* (*Lactococcus lactis*) overgrew *Proteobacteria*, while in sample A this effect was not observed, as *Firmicutes* (*Streptococcus gallolyticus*) did not overpass (*Proteobacteria*). 

Sample B was in all cases more diverse than sample A in the number of bacterial families (in sample A: 45—0 h, 206—48 h, in sample B: 72—0 h, 261—48 h). Generally, 97–99% of the classified bacterial reads of both samples belonged to the 19 families summarized in Table 2 and Appendix A. The five most abundant families (their reads were detected in any sample at the minimal abundance over 4%), are summarized in Table 2. The other 14 bacterial families, which detected only <0.5% to 4%, are summarized in Appendix A and they were selected according to their frequency and the coverage of food-borne and important clinical pathogens within.

In both samples, gram-negative bacteria were the most dominant bacterial group. For both samples before cultivation (0 h), the most abundant were bacteria from the family *Enterobacteriaceae* (91% and 48%), but after cultivation (48 h) they decreased (to 54% and 16%). For both samples, it was caused by the growth of *Pseudomonadaceae* (from <0.5% to 15%, and from 25% to 44%), or for sample A also by the extensive growth of *Moraxellaceae* (genus *Acinetobacter*), which multiplied from a very low initial amount (<0.5% to 22%). Family *Moraxellaceae* (genus *Acinetobacter*) did not grow so high in sample B (only up to 6%), although it was detected in 0 h (5%). Generally, for sample A (mungo sprouts), the family *Enterobacteriaceae* remained the most abundant also after cultivation (54%), followed by *Moraxellaceae* (22%) and *Pseudomonadaceae* (15%). For sample B (bio-mixture), *Enterobacteriaceae* (16%) were overgrown by *Pseudomonadaceae* (44%) and they were comparable with *Streptococaceae* (also 16%), followed by a distance by *Moraxellaceae* (6%) and *Erwiniaceae* (4%). The abundancy of the family *Erwiniaceae* decreased (from 17% to 4%) during cultivation. The most abundant *Enterobacteriaceae* were *Escherichia* spp. and *Enterobacter* spp. for both samples, the most abundant *Erwiniaceae* it was *Pantoea* spp.

The only abundant gram-positive family was *Streptococcaceae*. In sample A (mungo sprouts), the family of *Streptococcaceae* did not propagate during cultivation as the reads number decreased from 8% to <0.5% and the dominant genus was *Streptococcus*. In sample B (bio-mixture), the family of *Streptococcaceae* propagated intensively during cultivation (from 1% to 16%), when the dominant genus was *Lactococcus*. 

Moreover, the species diversity of sprouts before cultivation is quite comparable to the findings of Margot et al. [26] for *Pseudomonas* (14.4%) (*Pseudomonas*: sample A < 0.5%, sample B 25%), *Enterobacter* (11.1%) (*Enterobacter*: in both samples 3%) and *Klebsiella* (10.3%) (*Klebsiella*: sample A 2%, sample B < 0.5%). In addition, Margot et al. abundantly detected *Janthinobacterium* (22%) (*Janthinobacterium*: <0.5% in both samples), but on the contrary Margot et al. did not detect for example *Lactococcus* spp.

Weiss et al. [40] characterized the cultivable microbiota of sprouts and claimed that *Pseudomonas* spp. can inhibit the growth of *Enterobacteriaceae*. This fact was observed in both samples in our study. The similar effect should be considered for *Lactococcus lactis*, which is dominant on the contrary to *Enterobacteriaceae* in sample B. Lactic acid bacteria can produce bacteriocins that display antimicrobial properties against other bacteria even closely related to the producer strain [41].

From the additional 14 families in Appendix A detected (selected from these detected only from <0.5% to 4% reads), 11 were gram-negative families (for example, *Yersiniaceae* and *Campylobacteriaceae*) and 3 were gram-positive families (*Leuconostocaceae* and *Enterococcaceae*). Most of them proved some growth during cultivation. The dominant genera for the family *Leuconostocaceae* were *Weisella* and *Leuconostoc*, for the family *Campylobacteraceae,* the genera *Campylobacter* and *Arcobacter*.

The presence of reads of *Campylobacter* spp. (3 resp., 34 reads) and *Salmonella* spp. (537, resp., 470 reads) corresponds to the fact that they may typically occur in animal manure (used as a fertilizer or contaminated water) [25,29,42].

### 3.2. Cultivation Methods and Determination of CFU/g

The results of the colony counts of the target microorganisms (CFU·g^−1^) are summarised in Table 3. We compared the increasing of colony counts (CFU·g^−1^) and classified reads (CR·g^−1^) after and before cultivation (30 °C, 48 h) as their logarithmic ratios. These increasements were approximately comparable for both samples in colony count on PCA and CSB (sample A: PCA—2.8 logs, CSB—2.7 logs, CR—2.0 logs, sample B: PCA—2.4 logs, CSB—2.1 logs, CR—2.1 logs) and *Enterobacteriaceae* count (sample A: VRBD—1.9 logs, CR—1.8 logs, sample B: VRBD—1.1 logs, CR—1.7 logs). The higher increasement in colony count than in classified reads was observed for sample A and *E. coli* (TBX—2.9 logs, CR—0.9 logs) and isolated lactic acid bacteria (MRS—3.4 logs, CR—0.5 logs), but lower for sample B in lactic acid bacteria (MRS—2.2 logs, CR—3.4 logs) and not determined for *E. coli* (TBX—not determined, CR—1.1 logs). 

Uncultivable dead cell DNA reads in the sample before cultivation could decrease the ratio of classified reads after and before cultivation in comparison to the colony count of cultivable cells. As for sample B, no typical (or even atypical) colonies of *E. coli* were detected on TBX for all used dilutions, it was possible to determinate their upper limit for CFU·g^−1^ (less than) despite its abundant reads by nanopore sequencing. Moreover, it is under discussion whether in sample B the growth of some target bacteria on selective media VRBD and TBX should not be decreased due to their lower fitness in the presence of highly abundant *Lactococus lactis*. 

As no colonies of coagulase positive staphylococci, presumptive *Bacillus cereus* or moulds or yeast were detected in all used dilutions of selective agar (BP agar, MYP agar and DRBC), it was possible only to determine their upper limit for CFU·g^−1^ (less than). It corresponds to the fact that all these target species were detected by nanopore sequencing with the abundancy below 0.5%. 

### 3.3. Comparison of MinION, Cultivation Methods and MALDI-TOF MS

From the agar plates of all selective media and all dilutions, different types of colonies were randomly sub isolated (in replicates according to their abundance), and isolates were lately identified by MALDI-TOF MS with the score value higher than two in most isolates (secure genus and probable species identification). The comparison of isolates identification by MALDI-TOF MS with data from NS (for the 20 most abundant species) is presented in Table 4 (for sample A) and in Table 5 (for sample B).

In total, it was obtained 119 isolates from sample A (mungo sprouts) (59 before, 60 after) and 107 isolates from sample B (bio mixture) (56 before, 51 after). The chosen agar media allowed to cover the bacteriome diversity. Only *Enterobacteriaceae* were isolated on TBX and VRBD, including some less plentiful such as *Citrobacter freundii* (in both samples) and *Klebsiella oxytoca* (in sample B), not isolated from any other agar medium. On the other hand, quite abundant *Pantoea agglomerans* (in sample B) was isolated only from CSB or PCA. MRS allowed to isolate the genera *Lactococcus*, *Streptococcus* and *Weisella*, when the last one grew only on MRS agar. PCA and CSB agar allowed to isolate different species of *Acinetobacter* spp. Spread plating on CSB and MYP allowed to isolate aerobic genera *Pseudomonas* (CSB agar) or *Aeromonas* (MYP agar). 

Generally, we were able to isolate the majority of the 20 most abundant species by NS (by reads). After sample cultivation, we did not isolate *Arcobacter butzleri* and *Comamonas testosteroni* (sample A) and *Pectobacterium carotovorum* (sample B), as the cultivation conditions were not favourable for their growth. In sample B, we did not isolate some less abundant species (*Kosakonia cowanii—*position 13, *Acinetobacter johnsonii*—position 17), but also more abundant *E. coli* (position 4). It is under discussion whether the high abundance of *L. lactis* in sample B did not decrease other bacteria ability to be cultivated (their fitness). On the other hand, some less abundant species were isolated, possibly due to the more distinct colony macromorphology, which increased the possibility of being sub isolated (e.g., in sample A: *Streptococcus gallolyticus*—position 59, in sample B: *Klebsiella variicola*—position 130). This comparison suggests that two main approaches should be considered to obtain as precise microbiome pictures as possible by cultivation methods. The first approach is to use selective media, which allow selective isolation of the supposed target microorganisms group, and to subisolate only some representative of all different types of colonies. The second approach, to be examinated in future experiments, is to use mainly nonselective media for fastidious microorganisms as well, but to subisolate and identify the statistically significant amount of all different types of colonies.

In both samples, some of the 20 most abundant species before cultivation (by reads) are not involved in Table 4 and Table 5, as after cultivation neither they were among the twenty most abundant species (by reads) not subisolated. It is under discussion whether their reads before cultivation corresponded to dead cell DNA or the cultivation conditions were not favourable for them. 

Comparing nanopore sequencing and MALDI-TOF MS identification showed the limits of these methods according to their principles. As nanopore sequencing is based on DNA analysis, it identifies more precisely and more recently due to more updated databases. MALDI-TOF MS identification is based on comparing the protein profile with these in the used database, which is the main limitation (Biotyper 3.1 contains 6903 spectra for 2461 species and 424 genera). The other limitation is the lower discrimination power of protein spectra of some very closely related species or species complexes (for example, *Enterobacter cloacae* complex, different *Pseudomonas* species complex). The limitation of MALDI-TOF MS due to the database, in which not all isolated microorganisms may be present, leads to the general impossible identification (if the genus is not covered) or not correct species identification (if only some species of the given genus are covered). In this study, we detected only incorrect species identification, as all bacterial isolates were identified at least on the genus level. For example, none of *Acinetobacter* spp. isolates in the sample A was identified as *Acinetobacter soli*, detected by nanopore sequencing, as this species is not included in Biotyper 3.1. On the other hand, some *Acinetobacter* spp. isolates from sample B, identified as *A*. *bereziniae* or *A. guillouiae* (sample B) by MALDI-TOF MS (but only on the level of probable species identification level), were not detected by nanopore sequencing, as they correspond to some other detected *Acinetobacter* (e.g., *Acinetobacter equi*, *oleivorans*) species, not covered by Biotyper 3.1. 

The comparison of NS and cultivation methods showed that the NS bacteriome picture is more complex and more precise and it also enables the microorganisms semiquantification.

Generally, NS furnishes in one reaction such a complex picture that would be achieved only by using many different cultivation methods with a broad range of used media and cultivation conditions. NS also allows to detect microorganisms that could not be cultivated due to unsuitable conditions. Therefore, for obtaining the complex microbiome, NS is even less expansive than cultivation methods. However, the cultivation methods (following MALDI-TOF MS identification) enable to obtain individual bacterial isolates, which can be examined even for ARGs presence. This examination can be used for a deeper interpretation of NS results.

The weak point, especially for the semiquantification, is that the NS protocol did not allow to distinguish between DNA from alive and dead cells. It requires another investigation whether and how it should be possible to use ethidium monoazide (EMA) or propidium monoazide (PMA) dyes before DNA isolation, which method is established in PCR methods to avoid amplification of dead cell DNA [43].

### 3.4. Species Diversity by MinION

In sample A (48 h), the most dominant bacterial species by MinION reads were *Acinetobacter* spp. (19%) (*A. baylii*, *A. baumannii*, *A. johnsonii* and *A. soli*); *Enterobacter* spp. (17%) (*E. cloacae* and other species strains of *E. cloacae* complex; *Klebsiella* spp. (11%) (*K. pneumoniae* and *variicola*); *E. coli* (7%) and *Pseudomonas* spp. (8%) (*P. putida*). In sample A before cultivation (0 h), the most dominant bacterial genera were *Escherichia coli* (6689 reads), followed by *Streptococcus gallolyticus* (341 reads), *Enterobacter cloacae* (95 reads) and *Klebsiella pneumoniae* (84 reads). It is evidential that cultivation may have different effects on single species and partially change their abundancy. As some species detected before cultivation multiplied, the other species did not (as the cultivation conditions were not favorable or DNA could be detected from their dead cells). On the other hand, the cultivation allowed to multiply those species, in which the number of reads before cultivation was below the method detection limit. For example, *Acinetobacter* sp. ADP1 (*A. baylii* ADP1) reads increased by 4 logs, which moved it from position 35 to 1. On the other hand, *Pseudomonas putida*, not detected before cultivation, multiplied by 4.1 logs, and moved to position 5 after cultivation. The cultivation did almost not increase the reads of *Streptococcus gallolyticus* reads and it dropped from position 2 to 59. The frequency positions for *Enterobacteriaceae* such as *Escherichia coli, Klebsiella pneumoniae* and *Enterobacter cloacae* did not change significantly as the number of reads grew rapidly after cultivation. 

In sample B (48 h), the most dominant bacterial genera were *Pseudomonas* spp. (23%), *Lactococcus* spp. (16%) and species *Escherichia coli* (4%). From the genus *Pseudomonas* spp., species such as *Pseudomonas fluorescens*, *P. putida* and *P. alkylphenolica* were the most common. The second most frequent genus was *Lactococcus* spp. with the most identified species such as *Lactococcus lactis*, *L. raffinolactis* and *L. garvieae*.

In sample B, the cultivation did not change the frequency positions for *Escherichia coli* (reads before cultivation) (4373), followed by *Pantoea agglomerans* (1491) and *Pseudomonas fluorescens* (1224), for which the ranking positions changed maximally by 5 ranks. However, the highest increase of reads after cultivation was observed for *Lactococcus lactis* by 3 logs (from position 31 to 1), *Pseudomonas putida* by 3 logs (from position 20 to 3) and *Pseudomonas fluorescens* by 1 log (from position 3 to 2). The changes in the species composition during cultivation are based on the initial concentration of viable cells, the cultivation conditions including nutrient spectrum, the interspecies relations, and changes during cultivation.

### 3.5. Resistome

Immediately before cultivation, it was identified only 7 ARGs with 10 alignments (max. 2 alignments per gene, average 1.4) in sample A and 42 ARGs with 96 alignments (max. 13 alignments per gene, average 2.3) in sample B. The other present ARGs were not detected as their abundancy was below the method detection limit. However, after 48 h cultivation (30 °C) it increased to 236 ARGs with 5225 alignments in sample A and 192 ARGs with 4505 alignments in sample B (Table 6). All ARGs detected in the sample before cultivation were also present after cultivation. These ARGs reflect the cultivable resistome of these species, who multiply at the given cultivation conditions. If these species harbor some ARGs, which inhibit their growth, such ARGs will not be discovered under these conditions. The five dominant detected resistance mechanisms (Table 6) were efflux pumps, target alteration (for example, 16S rRNA gene mutation), enzymatic inactivation (production of beta-lactamases), reduced permeability (porin mutation), and target protection.

From the point of ARGs diversity, samples A and B differed not only in the amount of detected ARGs (236 and 192 ARGs), but also in the composition. In the sample A, most of the single detected ARGs were efflux pumps in gram-negative bacteria (39%), followed by target alteration (27%) and enzymatic inactivation (24%). For the target alteration, the most prevalent ARGs were 16S rRNA gene mutations, which inhibit mainly aminoglycosides from binding to the ribosome (12%), and mutations in topoisomerases *gyr*A, *gyr*B, *par*C, and *par*E, conferring resistance to fluoroquinolones (2%). Most ARGs of enzymatic inactivation were covered by genes for beta-lactamases (22%). Reduced permeability and target protection are present in 2% and 1% of all kinds of ARGs. The other mechanisms (7%) comprise mainly ARGs, coding different regulatory proteins or their mutations (e.g., positive regulators for increasing the transcription of efflux pumps). 

Contrary, in sample B, the ARGs for target alteration surmounted this encoding enzymatic inactivation. The target alteration was the most represented (41%), followed by efflux pumps (36%) and enzymatic inactivation (17%). In addition, also in sample B, the most prevalent ARGs variants for the target alteration were different variants of mutations of 16S rRNA (21%) and mutations in topoisomerases *gyr*A, *gyr*B, *par*C, and *par*E (3%) and for the enzymatic inactivation beta-lactamases (17%). Mechanisms for reduced permeability and target protection covered 2% of ARGs in both cases. The representation of the other mechanisms was lower (3%).

However, the diversity of ARGs of single mechanisms does not reflect their abundancy. The alignment number of ARGs depends on the ability of the host bacterium to grow and its initial amount of cells. Most ARGs alignments in both samples belonged to efflux pumps (32%, resp., 31%) (chromosomally or plasmid encoded) or 16S rRNA gene mutations (chromosomally encoded) (29%, resp., 41%). 

The other abundant ARGs by alignments were mutations in topoisomerase (for example, *Salmonella enterica gyr*A conferring resistance to fluoroquinolones, ARO: 3003926) (6%, resp. 5%), EF-Tu mutation (for example, *Escherichia coli* EF-Tu mutants conferring resistance to pulvomycin, ARO: 3003369) (8%, resp., 9%) and other mechanisms (such as regulator proteins—*mgr*B, ARO: 3003820) (5% in both samples).

The diversity of ARGs for enzymatic inactivation did not correspond to their frequency, as mainly for beta-lactamases in gram-negative bacteria (classes OXA, SHV, CTX-M, ACT and others), which could be also encoded in plasmids. In both samples before cultivation, genes for beta-lactamases were not detected. After cultivation different beta-lactamases were abundantly present mainly in *Enterobacteriaceae* (sample A—53 genes, sample B—32 genes), but with maximally 10 copies. The family *Enterobacteriaceae*, especially *Escherichia coli* and *Klebsiella pneumoniae* are mainly producers of an extended spectrum of beta-lactamases (ESBL), which can cause serious nosocomial infections [46]. The most clinically important ESBL families are TEM, SHV, and CTX-M [47]. Kim et al. analysed 91 samples of different sprouts and they isolated ESBL positive *E. coli* in 3.3% and ESBL positive *K. pneumoniae* in 16.5% of samples [48]. These isolates harboured *bla*_TEM_, *bla*_SHV_ and *bla*_CTX-M_ genes or their combination. Margot et al. analysed 102 samples of different sprouts from retail and three samples (2.9%) were found to be positive for ESBL. They isolated *K. variicola* and *E. coli* harboured *bla*_CTX-M-14_ and *Enterobacter cloacae* harboured *bla*_CTX-M-3_) [49]. Compared to our results, we detected by nanopore sequencing the genes *bla*_SHV_ and *bla*_CTX-M_ in *K. pneumoniae* in sample A after cultivation. This species assignment should be later confirmed by other methods (as e.g., PCR protocols).

In both samples, the 16S rRNA mutations were dominantly represented by three ARGs: a mutation conferring resistance to spectinomycin described in *Salmonella enterica* serovar Typhimurium (11%, resp., 14% alignments) (ARO: 3003512), mutations in the *rrs*H gene conferring resistance to spectinomycin described in *E. coli* (4%, resp., 6% alignments) (ARO: 3003372) and mutation conferring resistance to spectinomycin in *Neisseria meningitidis* (ARO: 3003497) covered 3% of them for both samples. 

For efflux pumps, 79 genes (sample A) and 63 genes (sample B) were detected, which were spread across the present gram-negative species (such as *Acinetobacter*, *Pseudomonas* or *Enterobacteriaceae*) and in sample B also in *Lactococcus lactis*. The most dominant genes were mainly encoding the transporters of resistance-nodulation-cell division (RND) antibiotic efflux pumps such as *mex*K (ARO: 3003693), *mex*F (ARO: 3000804), *mex*B (ARO: 3000378) or *tri*C (ARO: 3003681) in sample A, or *acr*B (ARO: 3000216) and *acr*D (ARO: 3000491) in sample B or these encoding ATP-binding cassette (ABC) antibiotic efflux pump as *yoj*1 (ARO: 3003952) in sample B.

Furthermore, the distribution of ARGs within the species was determined (Table 7). This derives from the alignment of the obtained sequences against ARGs database, which includes ARGs as unique sequences (from NCBI database) discovered in a particular species. For this reason, the ARGs alignment to a single species should be highly influenced by the database coverage of its ARGs and less precise, if compared to the species abundancy in sample. However, it seems that ARGs alignment to a genus or a higher taxonomic group level overpasses this discrepancy. However, generally from 236 ARGs in the sample A, almost 94% were assigned to gram-negative bacteria and 6% to gram-positive bacteria. Contrary, in sample B, from the 192 ARGs, almost 84% were assigned to gram-negative bacteria and 16% to gram-positive bacteria, which is comparable to the higher proportion of gram-positive bacteria (*Lactococcus* spp.). 

In sample A, there were assigned 40% of the detected genes to *Escherichia coli* (53% of alignments, mainly *Escherichia coli* 16S rRNA mutation in the *rrs*H gene conferring resistance to spectinomycin), 10% to *Pseudomonas aeruginosa* (8% of alignments), 9% to *Acinetobacter baumannii* (3% of alignments), 8% to the *Klebsiella pneumoniae* (4% of alignments), 13% to other gram-negative bacteria (4% to alignments), 3% to the *Mycobacterium* spp. (<0.5% of alignments), and 3% to other gram-positive bacteria (<0.5% of alignments). 

In sample B, there were assigned 34% of the detected genes to *Escherichia coli* (38% of alignments), 14% to *Pseudomonas aeruginosa* (26%), 5% to the *Acinetobacter baumannii* (1% of alignments), 4% to the *Klebsiella pneumoniae* (1%), 17% to other gram-negative bacteria (3%), 9% to the *Mycobacterium* spp. (2%) and 1% to other gram-positive bacteria (1%). 

Although single ARGs can harbour more species (within one family or in different families), the CARD database, on which the comparison with EPI2ME runs, presents a single ARG as one of its specific NCBI sequence, isolated from one concrete species. However, CARD also offers information about the prevalence of ARGs in the other important species, which can give a more precise picture together with the information about the species abundancy in samples. For example, *gyr*B mutations determined to *Salmonella enterica* are known to be present also in other *Enterobacteriaceae* (e.g., in *Citrobacter freundii* and *Klebsiella pneumoniae*). Another example is a large assignment of ARGs to *Pseudomonas aeruginosa,* which was detected only in low reads. It is under discussion whether at least some ARGs were not harboured by far more frequent species as *Pseudomonas putida* or *Pseudomonas fluorescens*. 

NS drew the picture of ARGs in both samples except the exact species assignment of a single ARG, which is the limit of EPI2ME analysis. Analysing the obtained isolates for the presence of detected ARGs could improve this given picture and overpass this limit. Despite this partial limit, NS seems to be a powerful tool for different analyses of resistome diversity and changes according to the environmental conditions (e.g., sublethal presence of antimicrobial compounds). 

## 4. Conclusions

Two samples of bean sprouts from a retail market (Czech Republic, 2020) were analyzed by nanopore sequencing to obtain information about bacteriome diversity and ARGs distribution (including resistance mechanisms and species occurrence). The results for bacteriome diversity were compared with standard cultivation methods and MALDI-TOF MS.

The most dominant families were *Enterobacteriaceae*, *Pseudomonadaceae*, *Streptococcaceae*, *Erwiniaceae* and *Moracellaceae*. Generally, in both samples, cultivation (30 °C, 48 h) decreased the abundancy of *Enterobacteriaceae* and increased the abundancy of *Pseudomonadaceae* or *Streptococcaceae* (only in sample B, presence of *Lactococcus lactis*). With the usage of suitable isolation techniques, it would be also possible to analyze viral or eukaryotic DNA.

MALDI-TOF MS identification of isolates obtained by cultivation methods corresponded to nanopore sequencing results, except for a few species not covered by the used database Biotyper 3.1. 

ARGs were dominantly harbored by the present gram-negative bacteria in both samples. The species assignment of most ARGs to *Escherichia coli* and almost rare *Pseudomonas aeruginosa* reflects their position of highly described species in CARD from *Enterobacteriaceae* and *Pseudomonas* spp. Most of the detected ARGs alignments encoded efflux pumps and target alteration (for example, 16S rRNA gene mutation), while ARGs encoding enzymatic inactivation (production of beta-lactamases), reduced permeability (porin mutation) and target protection were less abundant. We proved that nanopore sequencing is a rapid and efficient method for the determination of bacteriome and resistome of food samples.

## Figures and Tables

**Table 1 microorganisms-09-00937-t001:** Overview of reads obtained by nanopore sequencing for sample A and B after and before cultivation.

Reads	Sample A (0 h)	Sample A (48 h)	Sample B (0 h)	Sample B (48 h)
Analyzed	14,500	248,000	24,000	584,000
Reads·g^−1^ (analyzed)	1.4 × 10^6^	9.6 × 10^7^	1.5 × 10^6^	1.6 × 10^8^
Unclassified	14%	15%	43%	35%
Classified all	86%	85%	57%	65%
Classified only to root	29%	2%	10%	2%
Classified at least to genus	57%	83%	47%	63%
Bacteria	97%	99%	97%	99%
Reads·g^−1^ (classified bacteria)	7.6 × 10^5^	7.9 × 10^7^	6.9 × 10^5^	9.8 × 10^7^
*Proteobacteria*	91.9%	99%	97%	78.8%
*Firmicutes*	7.9%	0.6%	2.6%	20.6%
*Bacteroidetes*	<0.1%	<0.1%	<0.1%	<0.2%
*Actinobacteria*	0%	0.1%	<0.2%	<0.2%
Archaea	<1%	<1%	<1%	<1%
Viruses	1%	<1%	<1%	<1%
Eukaryota	1%	1%	3%	1%

**Table 2 microorganisms-09-00937-t002:** Family diversity of bean sprouts: the five most abundant families (detected in any sample with abundance reads over 4%).

Family/Genus	Sample A (0 h) 8059	Sample A (48 h) 202,867	Sample B (0 h) 10,807	Sample B (48 h) 366,947
***Enterobacteriaceae* (G−)**	**7312 (91%)**	**108,560 (54%)**	**5145 (48%)**	**58,346 (16%)**
*Escherichia*	83%	7%	40%	4%
*Enterobacter*	3%	22%	3%	7%
*Kosakonia*	<0.5%	1%	1%	1%
*Klebsiella*	2%	15%	<0.5%	1%
*Citrobacter*	<0.5%	4%	<0.5%	1%
*Cronobacter*	<0.5%	1%	<0.5%	<0.5%
*Raoultella*	<0.5%	2%	<0.5%	<0.5%
*Salmonella*	<0.5%	<0.5%	<0.5%	<0.5%
***Pseudomonadaceae* (G−)**	**3 (<0.5%)**	**30,577 (15%)**	**2720 (25%)**	**161,178 (44%)**
*Pseudomonas*	<0.5%	15%	25%	44%
***Streptococcaceae* (G+)**	**629 (8%)**	**492 (<0.5%)**	**63 (1%)**	**59,049 (16%)**
*Streptococcus*	8%	<0.5%	<0.5%	<0.5%
*Lactococcus*	0%	<0.5%	1%	16%
***Erwiniaceae* (G−)**	**4 (<0.5%)**	**347 (<0.5%)**	**1802 (17%)**	**14,142 (4%)**
*Erwinia*	<0.5%	<0.5%	2%	1%
*Pantoea*	<0.5%	<0.5%	15%	3%
***Moraxellaceae* (G−)**	**30 (<0.5%)**	**44,203 (22%)**	**534 (5%)**	**21,598 (6%)**
*Acinetobacter*	<0.5%	22%	5%	6%
Total (19 abundant families)	99%	98%	98%	97%
Gram-negative families	91%	97%	96%	77%
Gram-positive families	8%	<0.5%	2%	20%

**Table 3 microorganisms-09-00937-t003:** Colony count of target microorganisms.

Method (ISO)	Agar	Target Microorganism	Sample A (0 h) (CFU·g^−1^)	Sample A (48 h) (CFU·g^−1^)	Sample B (0 h) (CFU·g^−1^)	Sample B (48 h) (CFU·g^−1^)
4833	PCA	Colony count at 30 °C	8.7 × 10^7^	5.9 × 10^10^	5.9 × 10^8^	1.5 × 10^11^
-	CSB	Colony count at 30 °C	3.8 × 10^7^	2.0 × 10^10^	7.0 × 10^8^	1.0 × 10^11^
21528-2	VRBD	*Enterobacteriaceae*	1.2 × 10^7^	9.0 × 10^8^	1.2 × 10^8^	1.6 × 10^9^
16649-2	TBX	*Escherichia coli*	1.9 × 10^5^	1.4 × 10^8^	<1.0 × 10^3^	<1.0 × 10^4^
15214	MRS	Lactic acid bacteria	7.1 × 10^5^	1.6 × 10^9^	3.0 × 10^7^	5.0 × 10^9^
6888-1	BP	Coagulase positive staphylococci	<1.0 × 10^2^	<1.0 × 10^7^	<1.0 × 10^4^	<1.0 × 10^3^
7932	MYP	Presumptive *Bacillus cereus*	<1.0 × 10^2^	<1.0 × 10^7^	<1.0 × 10^4^	<1.0 × 10^6^
21527	DRBC	Yeast and mould	<1.0 × 10^2^	<1.0 × 10^7^	<1.0 × 10^4^	<1.0 × 10^4^

**Table 4 microorganisms-09-00937-t004:** Sample A: Comparison of MinION and MALDI-TOF MS, ^a^—*Acinetobacter baylyi* ADP1 [44], ^b^—*Enterobacter roggenkampii*, ^c^—*Enterobacter hormaechei* subsp. *hoffmannii* [45], ND—not defined.

Taxon	Specifics of MALDI-TOF MS Identification	Reads		Position by Reads	
		48 h	0 h	48 h	0 h
*Acinetobacter* sp. ADP1 ^a^	*Acinetobacter baylyi*	21,635	2	1	35
*Acinetobacter baumannii*	*Acinetobacter baumannii*	13,329	16	6	9
*Acinetobacter johnsonii*	*Acinetobacter johnsonii*	2889	1	14	63
*Acinetobacter soli*	Not present in Biotyper 3.1	1182	0	19	ND
*Enterobacter cloacae*	*Enterobacter cloacae*	20,766	95	2	3
*E. cloacae* complex ‘Hoffmann cluster IV’ ^b^	*Enterobacter cloacae* complex	3298	9	11	11
*E. cloacae* complex sp. 35734		3145	9	12	12
*Enterobacter ludwigii*		2240	3	15	22
*Enterobacter asburiae*		2964	14	13	10
*E. cloacae* complex ‘Hoffmann cluster III’ ^c^		1458	1	18	38
*Klebsiella pneumoniae*	*Klebsiella pneumoniae*	17,266	84	3	4
*Klebsiella variicola*	*Klebsiella variicola*	5058	6	7	16
*Escherichia coli*	*Escherichia coli*	14,722	6689	4	1
*Pseudomonas putida*	*Pseudomonas putida* complex	13,492	0	5	ND
*Pseudomonas* sp. JY-G	*Pseudomonas* spp.	2157	0	16	ND
*Citrobacter freundii*	*Citrobacter freundii*	4455	9	8	13
*Citrobacter* sp. FDAARGOS_156	Not present in Biotyper 3.1	1563	2	17	31
*Raoultella ornithinolytica*	*Raoultella ornithinolytica*	3632	4	9	19
*Arcobacter butzleri*	NOT ISOLATED	3542	0	10	ND
*Comamonas testosteroni*	NOT ISOLATED	1128	0	20	ND
*Cronobacter sakazakii*	*Cronobacter* sp.	607	7	35	14
*Streptococcus gallolyticus*	*Streptococcus gallolyticus*	275	341	59	2

**Table 5 microorganisms-09-00937-t005:** Sample B: Comparison of MinION and MALDI-TOF MS identification, ND—not defined.

Taxon	Specifics of MALDI-TOF MS Identification	Reads		Position by Reads	
		48 h	0 h	48 h	0 h
*Lactococcus lactis*	*Lactococcus lactis*	40,430	20	1	31
*Lactococcus raffinolactis*	*Lactococcus raffinolactis*	14,239	38	5	24
*Lactococcus garvieae*	NOT ISOLATED	3250	2	16	150
*Pseudomonas fluorescens*	*Pseudomonas* spp.	27,860	1224	2	3
*Pseudomonas putida*	*Pseudomonas* spp.	22,926	40	3	20
*Pseudomonas alkylphenolica*		13,026	2	6	138
*Pseudomonas chlororaphis*		5581	30	10	26
*Pseudomonas mosselii*		4980	9	11	54
*Pseudomonas* sp. HLS-6		2970	0	19	0
*Pseudomonas protegens*		2739	19	20	33
*Pseudomonas monteilii*		4198	62	15	14
*Escherichia coli*	NOT ISOLATED	14,569	4373	4	1
*Pantoea agglomerans*	*Pantoea agglomerans*	10,543	1491	7	2
*Weissella cibaria*	*Weissella cibaria*	10,120	149	8	8
*Enterobacter asburiae*	*Enterobacter asburiae*	5890	70	9	12
*Enterobacter cloacae*	*Enterobacter cloacae*	3122	69	18	13
*Serratia liquefaciens*	*Serratia liquefaciens*	4546	42	12	18
*Kosakonia cowanii*	NOT ISOLATED	4273	150	13	7
*Pectobacterium carotovorum*	NOT ISOLATED	4210	10	14	47
*Acinetobacter johnsonii*	NOT ISOLATED	3237	209	17	5
*Acinetobacter pittii*	*Acinetobacter pittii*	1686	29	38	27
*Aeromonas hydrophila*	*Aeromonas* spp.	2111	0	33	0
*Erwinia gerundensis*	*Erwinia persicina*	1621	132	39	10
*Klebsiella pneumoniae*	NOT ISOLATED	673	9	68	50
*Klebsiella variicola*	*Klebsiella variicola*	188	0	130	0
*Klebsiella oxytoca*	*Klebsiella oxytoca*	998	5	54	76
*Citrobacter freundii*	*Citrobacter freundii*	1203	2	45	112
*Cronobacter sakazakii*	*Cronobacter* spp.	65	4	173	88
*Paenibacillus naphthalenovorans*	*Paenibacillus naphthalenovorans*	1	0	1685	ND
*Acinetobacter* spp. not covered by Biotyper 3.1	*Acinetobacter bereziniae*/*quillouiae*	4829	99	ND	ND

**Table 6 microorganisms-09-00937-t006:** Representation of individual mechanisms of resistance across detected ARGs, ^a^—resistance to aminoglycoside, ^b^—resistance to fluoroquinolones, ^c^—resistance to elfamycin.

Resistance Mechanisms	Sample A (48 h)		Sample B (48 h)	
	Genes	Alignments	Genes	Alignments
Total number	236	5225	192	4505
Efflux pumps	39%	32%	36%	31%
Target alteration	27%	51%	41%	59%
16S rRNA mutation ^a^	12%	29%	21%	41%
Mutations in *gyr*A, *gyr*B, *par*C, *par*E ^b^	2%	6%	3%	5%
EF-Tu mutation ^c^	7%	8%	7%	9%
Other mutations (point mutation)	4%	5%	6%	3%
Enzymatic inactivation	24%	3%	17%	1%
Beta-lactamase	22%	2%	14%	1%
Enzymatic modification	2%	1%	3%	<0.5%
Reduced permeability—mutant forms of the porin Omp36	2%	4%	2%	1%
Target protection	1%	2%	2%	1%
Other mechanisms	7%	9%	3%	7%

**Table 7 microorganisms-09-00937-t007:** Distribution of ARGs through species, ^a^—reads of single species.

Taxon	Sample A (48h)			Sample B (48h)		
	Genes	Alignments	Reads ^a^	Genes	Alignments	Reads ^a^
Total number	236	5225	202,867	192	4505	366,947
Gram-negative bacteria	94%	99%	44%	84%	94%	15%
*Escherichia coli*	40%	53%	7%	34%	38%	4%
*Enterobacter cloacae*	6%	6%	10%	5%	2%	1%
*Klebsiella pneumoniae*	8%	4%	9%	4%	1%	<0.5%
*Acinetobacter baumannii*	9%	3%	7%	5%	1%	1%
*Pseudomonas aeruginosa*	10%	8%	<0.5%	14%	26%	<0.5%
*Salmonella enterica*	4%	17%	<0.5%	4%	18%	<0.5%
*Neisseria gonorrhoeae*	1%	1%	<0.5%	1%	2%	<0.5%
*Neisseria meningitidis*	<0.5%	3%	<0.5%	1%	3%	<0.5%
*Stenotrophomonas maltophilia*	2%	1%	<0.5%	1%	<0.5%	<0.5
Other gram-negative bacteria	13%	4%	11%	17%	3%	9%
Gram-positive bacteria	6%	1%	<0.5%	16%	6%	11%
*Enterococcus faecium*	<0.5%	1%	<0.5%	1%	1%	<0.5%
*Lactococcus lactis*	0%	0%	<0.5%	1%	2%	11%
*Mycobacterium* spp.	3%	<0.5%	<0.5%	9%	2%	<0.5%
Other gram-positive bacteria	3%	<0.5%	<0.5%	4%	1%	<0.5%

## Data Availability

Data are available on request to the corresponding author.

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
