# Peer review of "Application of Nanopore Sequencing (MinION) for the Analysis of Bacteriome and Resistome of Bean Sprouts"

_microorganisms, 2021, doi:10.3390/microorganisms9050937_

Round 1

Reviewer 1 Report

File attached

Reviewer 2 Report

Rapid methods for the parallel analysis of bacteriomas and resistance in food samples to increase food safety and prevent the spread of antibiotic resistance genes is an important issue globally. In this work, nanopore sequencing was used to analyze the diversity and dynamics of the microbiome as well as the resistance of two bean sprouts samples from the Czech retail market. The presented article is interesting, but it contains some inaccuracies that need to be cleared up.

The keywords repeat the title of the manuscript. They must be changed to reflect the most important keys used when searching for the article in search engines.

The abstract does not fully reflect the background, purpose, methods and results obtained. Therefore, it needs to be improved. This improvement should indicate to the reader what the purpose of the research was and what the results of this research show. Also for practice.

The introduction is quite extensive, but does not provide enough information on the topic of the manuscript. Therefore, the authors are asked to specify the background for the conducted research in relation to the existing knowledge. In its current form, it does not indicate a research problem.
One question that raises doubts is the purposefulness of the choice of research material. The authors did not provide the reasons that led the authors to undertake such research. There is no specific research goal. The authors are asked to indicate the purpose of the experiment in two sentences.

The description of the results is. too long. It should be shortened. Some descriptions are repeated in the discussion. Perhaps it would be better if the authors considered combining these two chapters. In this way, they would avoid unnecessary repetition and discussion of the research results twice.
Regarding rapid identification methods, please indicate at what time the entire study was carried out. This is especially important when a new, safe and rapid method of analyzing bacteriomas and resistance in food samples is proposed.
Please also indicate the applicability of this method for adaptation to other food samples, including processed food.

The summary is extensive. However, only one sentence is actually a concrete conclusion from this experiment, namely the information contained in the line: 727-728: ... "We proved that nanopore sequencing is a rapid and efficient method for determination of bacteriome and resistome of food sample." ... Therefore, the authors are asked to formulate the most important conclusions resulting from the research carried out.

Authors are requested to edit the literature in accordance with the journal's requirements.

Round 2

Reviewer 2 Report

The authors significantly improved the manuscript as recommended. It may be published in its current form.